A hybrid integration framework based on LOOCV and SARIMA: relationship exploring and predictive analysis between discipline attention and literature research

Zhao Yulin 1
Li Junke 2 ljk2006ljk@163.com
Liu Kai 2 3
Shang Chaowang 1
1 Faculty of Artificial Intelligence in Education, Central China Normal University , Wuhan , China
2 School of Information Engineering, Suqian University , Jiangsu , China
3 School of Computer and Information, Qiannan Normal University for Nationalities , Duyun , China
Kong Xiangjie
Electronic publication date: 2025 Apr 1
Publication date: 2025
Volume: 11
Electronic Location ID: e2754
Received 2024 Oct 29; Accepted 2025 Feb 19
Copyright: © 2025 Zhao et al.
Copyright year: 2025
Copyright holder: Zhao et al.
License: This is an open access article distributed under the terms of the Creative Commons Attribution License, which permits unrestricted use, distribution, reproduction and adaptation in any medium and for any purpose provided that it is properly attributed. For attribution, the original author(s), title, publication source (PeerJ Computer Science) and either DOI or URL of the article must be cited.
License URL: https://creativecommons.org/licenses/by/4.0/

Keywords: Discipline attention, Literature research, Relationship exploration, SARIMA, Predictive analysis

Funding: National Natural Science Foundation of China 62262055 Qing Lan Project of Jiangsu Province Suqian Talent Xiongying Project 2023-0035 High Level Talent Foundation of Suqian University 2024XRC011 Science and Technology Foundation of Guizhou Province [2019]1447 Sichuan Science and Technology Program 2023NSFSC1413, 2023YFG0117 This work was supported by the National Natural Science Foundation of China (Grant 62262055); Qing Lan Project of Jiangsu Province; Suqian Talent Xiongying Project (Grant No. 2023-0035); the High Level Talent Foundation of Suqian University (Grant No. 2024XRC011); the Science and Technology Foundation of Guizhou Province (Grant No. [2019]1447); the Sichuan Science and Technology Program (Grant Nos. 2023NSFSC1413, 2023YFG0117). There was no additional external funding received for this study. The funders had no role in study design, data collection and analysis, decision to publish, or preparation of the manuscript.

==============================
Analyzing the relationship between the discipline of network attention and literature research can provide new insights for the innovative development of future disciplines. Many current studies focus on network attention, but its innovative application in the field of subject teaching has not been fully verified. Based on this, this paper proposed a relationship analysis and predictive analysis (RAPA) framework based on leave-one-out cross-validation (LOOCV) and Seasonal Auto-Regressive Integrated Moving Average (SARIMA) to explore the relationship between subject attention and literature research from the perspective of junior high school information technology. Based on the RAPA framework, five key keywords of this subject were extracted by combining the Baidu Index and China National Knowledge Infrastructure (CNKI) in first. Secondly, LOOCV was used to explore the relationship between subject attention represented by keywords and literature researches. Then, SARIMA was used to predict the future trends of subject attention and its literature researches. Finally, the prediction errors of different methods were compared. Based on the RAPA framework, the correlation analysis found that the r-values of subject attention and literature researches were all greater than 0.75, indicating a positive correlation between them. The predictive analysis found that the subject attention of junior high school information technology will be flat or decline in the next 2 years. Meanwhile, the amount of literature in this discipline has decreased compared to previous years, with an average of approximately 136. The prediction comparison showed that the prediction method in this study has a smaller mean absolute error (MAE) than other methods, and the MAE difference is 3.51. This indicated that subject attention, as an auxiliary variable of scientific research literature, is conducive to the quantitative analysis of literature research. At the same time, this study revealed the influence and role of big data represented by Internet attention in educational research.

Introduction

As a subject in the basic education stage, junior high school information technology is based on the background of cultivating students’ learning psychology, discipline behavior, and learning methods of computers. It is a discipline that emphasizes both theory and practice. Subject network concern refers to the current scholars’ concern about the hot words and information of the subject. With the popularity of the Internet and intelligent devices, people are no longer limited to academic journals, and more and more people use computers and mobile terminals to search for relevant information. Compared with traditional sci-tech periodicals, network media has a short time lag, fast transmission speed, and more easily meets the needs of scholars. However, whether there is a complex dynamic relationship between academic attention based on network media and the amount of traditional journal literature is rarely proposed in the existing literature. As the media and carrier of knowledge dissemination, the research on the relationship between sci-tech periodical literature and network media is of practical significance.

In recent years, many scholars have been devoted to the study of online media attention and scientific and technological literature. Zhang et al. (2021) empirically analyzed the Google search index and found that Internet concern has a strong Granger causality with Bitcoin transaction volume. Based on the Baidu index, Xia & Hu (2021) revealed that investors’ concern about Corona Virus Disease 2019 (COVID-19) had a positive effect on the pharmaceutical stock market. Cascajares et al. (2021) found that environmental sciences and medicine are the most relevant fields in bibliometrics, second only to social sciences and computer science through bibliometric evaluation by Scopus and Web of Science (WoS). Some scholars have made a comparative analysis by using the literature research of scientific and technological journals and the hot spots of network concern. Huang (2019) proposed that tourism journals should be good at capturing the cutting-edge issues and publish them in time to improve their competitiveness by comparing the literature of tourism journals and the hot spots of Internet concern. Zhang et al. (2018) proposed a hot spot identification method for scientific research literature based on social network attention, and verified the timeliness of this method through multiple comparative experiments. The current research content is rich and diverse, providing reference resources for this study. However, most research has focused on single-dimensional analysis of online attention or scientific journals. For example, the impact of Internet attention on the market (Zhang et al., 2021; Xia & Hu, 2021), identifying research hotspots using online attention (Zhang et al., 2018), and bibliometric analysis of scientific journals (Cascajares et al., 2021). In addition, some studies have conducted comparative analysis between online attention and scientific journals, such as the comparative study on the hotspot timeliness of tourism journal literature and tourism attention (Huang, 2019). However, few studies have focused on analyzing the relationship between online attention and scientific journals, especially in the field of information technology that emphasizes the balance between theory and practice. In view of this, this article aims to explore the relationship between disciplinary attention and literature research from the perspective of information technology discipline. Under the guidance of this relationship, the number of articles in information technology discipline in the next 2 years is predicted.

To achieve this goal, a hybrid integration framework based on leave-one-out cross-validation (LOOCV) and Seasonal AutoRegressive Integrated Moving Average Model (SARIMA) is constructed. By analyzing the relationship between disciplinary attention and literature research, this framework further validates its effectiveness and strengthens the auxiliary role of disciplinary attention to literature research. The highlight of this study lies in the proposed hybrid integration framework RAPA. The RAPA framework is based on two dimensions: disciplinary attention and literature research, and integrates multiple methods such as Pearson Correlation, LOOCV, Multiple Regression, SARIMA, etc. In the RAPA framework, this article discusses several aspects of disciplinary attention and literature research in detail, including relationship analysis, prediction analysis, and prediction difference comparison. Therefore, this article’s research questions mainly focus on the following aspects:

RQ1. What is the relationship between subject network attention and its literature researches based on a new perspective of junior high school information technology?

RQ2. Based on the above relationship, what is the future trend of subject network attention and literature researches?

RQ3. Based on the prediction results, what is the discrepancy between our prediction method and other prediction methods?

In addition, the main contributions of this study are as follows:

(1) On the basis of previous studies on Internet attention (medicine, economy, agriculture, etc.), this article further extended the applied research on Internet attention in disciplines, so as to expand the research field of network attention.

(2) From the perspective of the information technology subject in junior high school, this article broke the previous review research on this subject and increased the quantitative research on the subject.

(3) This article proposed a hybrid integration framework relationship analysis and predictive analysis (RAPA) based on LOOCV and SARIMA, and confirmed the quantitative relationship between disciplinary attention and the amount of scientific literature from a disciplinary perspective. This provided a reference case for the quantitative analysis of scientific journal literature research.

Materials and Methods

Materials

The research object is from China National Knowledge Infrastructure (CNKI). Specifically, all the literature-themed “Junior High School Information Technology” from 2011 to 2023 was searched in Chinese Journal Full-text Database (CJFD), and keywords were extracted using keyword frequency analysis. This article first calculated the keyword frequency for each year using the complete counting method of word frequency analysis, then accumulated the word frequencies of the same keywords for all years and output them in reverse order. Finally, all keywords in the information technology subject of junior high school in the past decade were counted. These keywords were used as preliminary pre-selected objects of junior high school information technology subjects and further filtered as required.

The quantity of academic articles is indexed by the quantity of CNKI literature of the keywords collected. The subject network attention is measured by Baidu Index. In this article, network attention refers to the degree to which netizens pay attention to a certain social hot topic or thing in the era of the internet. When this kind of attention is specifically applied to a discipline, it forms the discipline network attention. The discipline network attention reflects the degree of people’s interest in research hotspots, article topics, or content related to the discipline. Usually, it is measured by the number of searches for relevant information about the subject, such as keywords, academic articles, etc. This indicator reflects the popularity and influence of the discipline on the Internet, and the Baidu Index is an important tool to measure this indicator. The Baidu Index is based on the record of massive search data of netizens on Baidu’s search platform, which can reflect social hot spots, and netizens’ interests and needs over a period of time. Specifically, Baidu Index is derived by weighting the search scale and frequency of keywords based on Baidu’s massive search behavior data of netizens. It provides a quantitative indicator of netizens’ attention and has been widely used by many scholars in various fields in recent years (Chen et al., 2023; Liu et al., 2024), with high authority and universality. Therefore, this article chooses Baidu index as the quantitative index of subject network attention.

This article collects the relevant data from the CNKI and Baidu Index, and analyzes the relationship between discipline literature researches and discipline network concern.

RAPA framework

According to the two dimensions of disciplinary attention and literature research, this article constructs a hybrid integration framework for relationship analysis and predictive analysis (RAPA) by using multiple methods comprehensively (Fig. 1). Meanwhile, this article verifies the effectiveness of the RAPA framework from the perspective of information technology discipline.

Figure 1 RAPA research framework.

Firstly, this article takes the discipline of information technology as an example and extracts the main keywords of the discipline by word frequency analysis. Then, based on the main keywords, it divides two dimensions: disciplinary attention and disciplinary literature research.

Secondly, according to the two dimensions divided, this study collects their respective datasets as the basic data source for the RAPA framework.

Finally, this article explores the specific application process of the RAPA framework using several methods such as Pearson Correlation, LOOCV, SARIMA, and Predictive regression, further verifying the effectiveness of the RAPA framework implementation.

The specific implementation process of the RAPA framework includes three steps, which correspond to three research questions RQ1, RQ2, and RQ3, respectively. Each step used a unique method for experimental data analysis, which is described in detail below.

Step 1: For RQ1, RAPA analyzed the relationship between subject network attention and subject literature researches.

Selecting representative data of subject network attention (Baidu Index) and CNKI subject literature researches, RAPA used the Pearson correlation coefficient to measure the relationship between subject attention and the number of its literature researches. The Pearson correlation coefficient can reflect the linear correlation between two variables and has been used in multiple studies (Alhawarat, Abdeljaber & Hilal, 2021). Therefore, this article referred to the Rahadian et al.’s (2023) practice and used quantitative data to analyze the correlation between them.

Step 2: For RQ2, RAPA predicted the future trend of subject network attention and subject literature researches.

Based on the above relationship, RAPA first established a regression model between subject attention and subject literature quantity by cross-validation method; secondly, using time series to predict subject attention; finally, the subject attention and the regression model were used to predict the subject literature quantity. The time series prediction method based on network attention has been applied to many fields (Wang et al., 2023; Yadav & Thakkar, 2024). This study drew on Luo et al.’s (2023) approach and predicted the future development trend of information technology in junior high school.

Step 3: For RQ3, RAPA explored the differences between different predictive methods.

Based on the prediction results of the subject literature researches, RAPA compared the prediction accuracy and differences between the method in this article and gray prediction (Aykaç Özen & Öbekcan, 2023) using two evaluation indicators, Mean Absolute Error (MAE) and Mean Absolute Percentage Error (MAPE). The comparison was used to verify the validity of the prediction method and the reliability of the experimental result in this article. The following was a detailed introduction to specific research methods.

Relationship analysis

Correlation calculation

In statistics, the linear Correlation between two variables A and B is generally expressed by the Pearson Correlation coefficient (r), whose value is between −1 and 1. The correlation coefficient can be defined only when the standard deviation of both variables is not zero. See Formula (1) for the expression. The greater the absolute value of r, the larger the relation between variables. The absolute value of r goes from 0 to 1, and the correlation goes from weak to strong.

(1) r=∑(xi−x¯)(yi−y¯)∑(xi−x¯)2∑(yi−y¯)2

LOOCV

As a statistical method for estimating the performance of machine learning models, Cross-validation (Fleming & Garen, 2022) generates multiple small training and test sets and uses these data sets to adjust the model parameters. The standard k-fold cross-validation is to cross-train the model with k-1 subsets and test the model with the other subsets. This method’s advantage is that all the sample data are used as training and verification models. LOOCV is a special cross-validation method (Zhao et al., 2023). In the training process of each iteration, one sample is used as the validation set, and the other n-1 samples are used as the training set. Repeat this process until each sample of the dataset is used as a verification set. Cross-validation helps optimize the model’s hyperparameters and improves its generalization performance through iterative training and validation on multiple subsets of data. On this basis, the article combines cross-validation to build a multiple regression equation, which is used to establish the relationship between discipline concern and the number of academic articles.

(2) lny=b+a1lnx1+a2lnx2+a3lnx3+......+anlnxn+ϵn.

In Formula (2), y represents the number of literature researches on the discipline. x1, x2,…, xn is the explanatory variable, namely the Baidu Index of multiple keywords reflecting the subject. b is the regression constant; a1, a2,…, an is the regression coefficient to be estimated, namely the mean value of model parameters trained by cross-validation. ε is the random error. In this article, both independent variables and dependent variables are logarithmically processed to eliminate heteroscedasticity.

Predictive analysis

SARIMA model

The SARIMA model is one of the predictive analysis methods of time series (Theerthagiri & Ruby, 2023). The general expression for the SARIMA model is:

(3) Φp(L)AP(Ls)ΔdΔsDyt=Θq(L)BQ(Ls)ϵt.

In Formula (3), P, Q, p, and q represent the maximum lag order of seasonal and non-seasonal autoregressions, and moving average operators respectively. D and d represent the differential order of seasonal and non-seasonal, respectively; s is the change period of seasonal series. The above formula can be denoted as ARIMA(p, d, q) × (P, D, Q)s-order seasonal time series model.

The main steps of SARIMA model construction and prediction are as follows:

(1) Test the stationarity of the original sequence. If the sequence is not stationary, transform the original sequence into a stable sequence by difference and seasonal difference, so that xt=ΔdΔsDyt;

(2) Build the model with xt, specifically:

a. Test the stationary sequence after difference, and use Akaike Information Criterion (AIC) and Bayesian Information Criterion (BIC) to determine the optimal order, and then determine the model parameters;

b. Establish a model with the estimated parameters, and perform model fitting to evaluate the model effectiveness.

c. Use the established model for forecasting.

Augmented Dickey-Fuller Test

The SARIMA model requires that the time series be stationary, meaning that the statistical patterns of the data do not change over time. Augmented Dickey-Fuller (ADF) test is a commonly used rigorous statistical test method (Zou & Politis, 2021). The calculation formula for ADF statistics is:

(4) ADF=(Yt−Yt−1)−δYt−1.

In Formula (4), Yt represents the current value of the time series, and Yt−1 represents the previous value of the time series. δ represents the trend of a time series, which can be a constant or a linear trend. If the value of the ADF statistic is less than a certain critical value, the null hypothesis can be rejected, indicating that the time series has a unit root. Otherwise, the null hypothesis cannot be rejected.

Prediction differences

MAE

MAE refers to the average absolute error between predicted values and actual values. The numerical unit of MAE is the same as the original data, which can intuitively reflect the size of prediction error. Its calculation formula is:

(5) MAE=1n∑i=1n|yi−y^i|

where n represents the number of samples, yi represents the actual value, and y^i represents the predicted value.

MAPE

MAPE not only considers the deviation between predicted values and actual values, but also the ratio between deviation and actual values. This indicator does not change due to the global scaling of the target variable (Barboza, Nunes Silva & Augusto Fiorucci, 2023). It is suitable for problems with large dimensional differences in the target variable. Its calculation formula is:

(6) MAPE=100%n∑i=1n|yi−y^iyi|.

The value of MAPE is a percentage, which can intuitively reflect the size of the prediction error relative to the actual value. In Formula (6), n represents the number of samples, yi represents the actual value, and y^i represents the predicted value.

Results

Relationship analysis and modeling

Based on RAPA framework, this article used correlation analysis, cross-validation and other methods to explore the relationship between subject network attention and subject literature researches.

Keyword screening

The screening of keywords follows the following steps:

(1) Extract Trending Keywords and their word frequency. Retrieve literature on the topic of “Junior High School Information Technology” from 2011 to 2023 on CNKI and export them. Then use VOSviewer to obtain Trending Keywords and their frequency.

(2) Construct a keyword frequency matrix. Count keywords and word frequencies by year, and put them in the list. In the list, each column represents each keyword, and each row represents the keyword frequency for each year, forming a word frequency matrix.

(3) Calculate the total frequency of each keyword. Count the total word frequency for all years based on the keywords in each column, and then sort them in reverse order according to the total word frequency. Table 1 shows the top 30 keywords in literature research on junior high school information technology.

Table 1 Trending Keywords about “middle school information technology” subject.

Num	Keywords	Frequency	Num	Keywords	Frequency	
1	Information technology	1,170	16	Information-based teaching	61	
2	Junior high school information technology	272	17	Teaching resources	57	
3	Junior high school mathematics	227	18	Educational informatization	55	
4	Teaching	221	19	Learning interest	48	
5	Classroom teaching	121	20	Teaching strategy	47	
6	Micro-lesson	119	21	Informatization environment	45	
7	Junior high school physics	117	22	Junior middle school mathematics teaching	40	
8	Modern information technology	113	23	Information literacy	39	
9	Junior high school information technology teaching	90	24	Multimedia technology	38	
10	Application	88	25	Instructional design	31	
11	Multimedia teaching	83	26	Junior high school biology	27	
12	Teaching mode	77	27	Micro video	27	
13	Flipped classroom	76	28	Autonomous learning	25	
14	Core literacy	69	29	Curriculum integration	22	
15	Efficient classroom	65	30	Computational thinking	21	

Relationship analysis

The preliminary extracted keywords have a large amount of data and complex information, so it is necessary to further streamline the sample to eliminate secondary words. Based on the experimental purpose, secondary filtration was conducted:

1. Remove words that are not relevant to the topic; such as: junior high school physics, junior high school biology;

2. Remove words that are not included in Baidu Index; such as: core literacy, computational thinking;

3. Remove words that cover a large range: such as: teaching, application.

Secondly, this article used Pearson correlation analysis to explore the relationship between subject attention and literature research represented by keywords. On this basis, keywords that are highly correlated with junior high school information technology subjects (r > 0.5) and have statistical significance are selected as research objects.

This article conducts correlation analysis on the filtered words through t-test. Meanwhile, based on the Pearson test results, select words that can characterize this subject and have statistical significance (p < 0.05): Information-based teaching (r = 0.887**), Micro-lesson (r = 0.841**), Flipped classroom (r = 0.798**), Multimedia teaching (r = 0.867**), Educational informatization (r = 0.753**). These keywords collectively reflect the key content and characteristics of junior high school information technology subject from the aspects of teaching mode, teaching resources, teaching methods, and teaching technology (Huang & Wang, 2020). Among them, information-based teaching emphasizes the use of information technology (such as computer, network, multimedia, etc.) to optimize the teaching process and improve the teaching effect. It reflects the modernization trend of teaching methods and technology in the field of junior high school information technology. Micro-lessons promote the fragmentation and personalization of teaching content, making teaching more flexible and efficient. It represents the innovation in the presentation of teaching content and teaching resources. Flipped classroom aims to fully utilize classroom time to enhance students’ self-learning and cooperation abilities (Qin & Lu, 2018). Its implementation process requires the support of information technology, which reflects the reform of the teaching mode of information technology. Multimedia teaching provides students with rich and diverse learning resources (such as images, sounds, videos, etc.), making the teaching content more vivid and intuitive. It reflects the diversity of teaching methods. Educational informatization is an important force in promoting the development of information technology disciplines, as well as a process of promoting educational modernization and reform. It emphasizes the comprehensive and profound impact of information technology on education. The selected keywords reflect the subject characteristics of middle school information technology from multiple dimensions, which together constitute the rich connotation of middle school information technology subject. In addition, it can be found that the discipline network attention represented by these five keywords is positively correlated with academic study (0.6 < r < 0.8 represents strongly related).

Relationship modeling

The simple correlation only represents the correlation between single factors, which cannot completely and objectively reflect the actual relationship between independent variable and the dependent variable. Therefore, multiple regression analysis is further applied based on simple correlation analysis (Duan, 2022). This article used the strong correlation reflected by keywords to build the relationship between the subject’s network attention with the number of academic articles. The network attention of the keywords screened in the previous step was selected as the independent variable, that is, the annual Baidu Index of information-based teaching, micro-lesson, flipped classroom, multimedia teaching, and education informatization. The number of academic articles on this subject is selected as the dependent variable, that is, the annual academic articles of CNKI-themed “Junior High School Information Technology”. The variable definitions are shown in Table 2.

Table 2 Model variables and their definitions.

Variable	Dimension	Indicator	Symbol	Data source	
Independent variable	Subject attention	Information-based teaching	x 1	Annual average of Baidu Index	
Micro-lesson	x 2	
Flipped classroom	x 3	
Multimedia teaching	x 4	
Educational informatization	x 5	
Dependent variable	Literature research on Junior high school information technology	The amount of literature on information technology topics in junior high school	y	Amount of CNKI annual publication	

The article selects the sample data set from 2011 to 2023 for model parameter training, and finally calculates the average of each regression coefficient. The regression equation obtained is:

(7) lny=0.366lnx1+0.345lnx2−0.149lnx3+0.655lnx4−0.143lnx5−0.483.

Formula (7) reflected the relationship between subject network attention and subject literature researches represented by keywords, and was also an answer to RQ1. Subject network attention refers to the attention paid by network users to subject keywords, which represents the development status of the subject. The subject literature researches are quantitative reflections of scholars’ studies on the subject. There was a positive correlation between the subject attention and the amount of literature represented by the keywords. Its relationship also reflected the dynamic correlation between the network attention effect and traditional scientific journals. This helped to predict the future publishing trend of this subject.

Predictive analysis

Based on RAPA framework, this article made a prediction analysis on the basis of the relationship between subject network attention and the number of academic papers. First, RAPA used the SARIMA model to predict the network attention of the keywords in the next 2 years based on historical observations. Then, according to the regression relationship model between the subject’s attention and its literature research, the publication number of the literature researches in this subject is further predicted.

Construction of predictive models

Based on the information technology subject of junior high school, this article predicts the subject attention according to the selected keywords. Considering the limited space, this article takes the keyword “Flipped classroom” as an example to analyze the construction process of the prediction model in detail. Other keywords are similar.

Sequence analysis

Figure 2 showed the original sequence of the ‘flipped classroom’, which exhibited a trend of regular fluctuations. By dividing the original time series into three parts: Trend, Seasonal, and Residual term, to observe the seasonal characteristics of the data. Figure 2 showed that the time series has a relatively obvious seasonal trend, with annual data showing a trend of first increasing and then decreasing. Meanwhile, the ADF test results of the original sequence were obtained (Table 3). Table 3 showed that p = 0.944730, much larger than 0.05, with unit roots. Therefore, the original sequence was non-stationary and required differential processing to meet the stationarity characteristics.

Figure 2 Time series decomposition of the model.

Table 3 ADF test results of the model.

Test statistic	p-value	Critical value (1%)	Critical value (5%)	Critical value (10%)	
−0.145348	0.944730	−3.510712	−2.896616	−2.585482	

Sequence stabilization and testing

From a statistical point of view, only stationary time series can avoid the existence of “pseudo-regression”. This article first performed first-order difference on the original sequence, followed by seasonal difference, and drew a test chart for the differential sequence (Fig. 3). Combining Fig. 3 and the ADF test, it was found that the mean and variance of the sequence after differential processing have stabilized. The p = 7.852714e−11, less than 0.05, indicating that the sequence is stable after one ordinary difference and one seasonal difference.

Figure 3 Sequence test after first-order difference of the model.

Model parameter determination

Model parameters can be determined by observing the trailing and truncating phenomenon of Autocorrelation Function (ACF) and Partial Autocorrelation Function (PACF) graphs. However, this method may not be entirely accurate and is subject to a certain degree of subjectivity. Therefore, it is necessary to combine multiple fitting models and select the appropriate model based on the AIC, BIC, and residual analysis results. Based on the maximum likelihood estimation method, this article used the auto_arima function in the pmdarima library for parameter estimation. Finally, the optimal model was selected according to AIC and BIC. The smaller the AIC and BIC, the better the fitting and prediction performance of the model.

In this case, ARIMA(1,1,1)(2,1,0)12 was selected as the optimal model according to the results of model operation and considering the significance of estimated parameters. By substituting the trained optimal model into Formula (3), the basic expression of the flipped classroom model was obtained:

(8) (1−φ1L)(1−α1L12−α2L24)ΔΔ12yt=(1+θ1L)ϵt.

By inputting the ARIMA(1,1,1)(2,1,0)12 model result parameters into Formula (8), the detailed expression of the model can be obtained as follows:

(9) (1−0.9237L)(1+0.702L12+0.5288L24)ΔΔ12yt=(1−0.5844L)ϵt

Model evaluation

Based on the estimated parameters, this article selected December 2019 to December 2023 for model evaluation and drew a model evaluation diagram (Fig. 4). According to Fig. 4, the trend of the fitted value and the actual value was basically consistent. The model has a good prediction effect. Therefore, this model can be used for subsequent prediction.

Figure 4 Comparison chart of the actual and fitted values of the model.

Similarly, build the model of Information-based teaching, Micro-lesson, Multimedia teaching and Educational informatization, and predict the future attention trend according to the modeling results.

Prediction of subject attention

Referring to the modeling and prediction process of the keyword “flipped classroom”, this article predicts other keywords to reflect the attention trend of junior high school information technology subjects (Fig. 5).

Figure 5 The prediction results of the model.

(A) Information-based teaching. (B) Micro-lesson. (C) Flipped classroom. (D) Multimedia teaching. (E) Educational informatization.

The prediction results of subject attention represented by keywords show that the hotspot of information technology subject will show a flat or declining trend in the next 2 years. Among them, information-based teaching, flipped classroom, and multimedia teaching show a stable trend, which is the same as the level of concern in recent years. However, Micro-lesson and educational informatization are showing a downward trend. Especially in the field of educational informatization, by 2025, its attention index will drop to around 175. The hotspot of information technology subject presents an unstable trend, and there has been a state differentiation. As a new type of teaching resources and methods, Multimedia teaching and flipped classrooms play an important role in integrating information technology and classroom teaching. It is also a hot topic that various disciplines will continue to pay attention to in the future (Qin & Lu, 2018). Since the rise of educational informatization, it has comprehensively promoted the use of modern information technology in the process of education and teaching. However, comparative analysis shows that China should continue to promote the construction of digital education infrastructure and improve the efficiency of digital education resources (Wang & Chen, 2022).

Prediction of the number of subject literature researches

The number of subject literature researches is obtained by substituting the predicted values of disciplinary attention into the regression equation. First, according to the relationship model between subject attention and literature research (Formula 7), the independent variable in 2024 was substituted into Formula (7) to obtain the dependent variable value. By further performing anti-logarithm treatment on the dependent variable, the number of literature researches on junior high school information technology in 2024 could be obtained, which was 145.

Secondly, to reduce prediction error, data from 2024 was added to the sample set and a regression prediction model with LOOCV was constructed. The resulting model is as follows:

(10) lny=−0.753+0.361x1+0.286x2−0.108x3+0.701x4−0.109x5.

Then, the independent variable of subject attention in 2025 was substituted into Formula (10). Further anti-logarithm treatment could obtain the number of literature researches in 2025, which was 123. Figure 6 shows the publication number of literature research in this subject over the next 2 years. It can be seen that the overall number of publications in junior high school information technology literature has shown a fluctuating trend over the years. After 2023, the publication number of academic literature tends to be flat. The number of publications has decreased compared with previous years, with an average of 136 articles.

Figure 6 Prediction results of literature researches.

The horizontal axis represents time, and the vertical axis represents the number of literature researches in this discipline.

An overall analysis of RQ2 found that the discipline attention represented by the five keywords presented a steady or declining trend. In recent years, the public’s attention to information technology subject in junior high school is not high, showing a slowing trend. Meanwhile, its academic literature researches would not increase in the next 2 years. Information technology subject is a compulsory course in primary and secondary schools, which is of significance to the reform of classroom teaching methods. Properly increasing literature researches of this discipline can effectively tap the key points within the discipline and promote the in-depth integration of multiple disciplines.

Prediction comparison

Based on the relationship between subject attention and literature research established in RAPA framework, this article predicted the future publication number of information technology subject themes in junior high school. To evaluate the prediction effect and ensure the necessity of establishing the above relational model, RAPA also used GM(1,1) (Althobaiti & Shabri, 2022; Yeh & Chang, 2021) to forecast the number of articles in this discipline. The two prediction methods were compared and analyzed to solve RQ3. In addition, RAPA selected the Standard Deviation (SD), MAE and MAPE of the prediction results as evaluation indicators for prediction accuracy. Table 4 listed the index values of GM(1,1) and attention-based regression prediction methods.

Table 4 Comparison of model accuracy.

Evaluating indicator	SD	MAE	MAPE	
GM(1,1)	35.12	37.78	14.35%	
Multiple regression prediction	30.92	34.27	13.21%	

According to the overall prediction accuracy in Table 4, the multiple regression prediction error combined with disciplinary attention was relatively small. It indicated that the method of using subject attention to predict literature quantity was of high accuracy. It also showed that network concern can assist discipline literature research and improve the accuracy of prediction. Specifically, the standard deviation of multiple regression prediction results was small, indicating that the forecast value was nearly the actual value, and the prediction result was relatively well. According to the calculated prediction results, there was little difference between the data results of the two methods, which also confirmed the correlation between the subject network attention reflected by the Baidu index and the amount of CNKI academic articles. Meanwhile, this also showed that it was reasonable to use the attention to predict the subject literature researches.

Discussion

The implications of the findings

This study aimed to explore the relationship between subject network attention and literature research from the perspective of information technology in junior high school. The main research results were summarized in three aspects, namely the correlation between disciplinary attention and literature research, the prediction results of disciplinary attention and its literature research, and the comparison of prediction accuracy. The results indicated a certain dynamic correlation between subject attention represented by the network index and the amount of literature research in scientific journals. The public’s behavioral willingness to pay attention to disciplines is recorded through big data, while their contribution to scientific research is reflected in the output of scientific journal literature. In recent years, the amount of information technology subject literature in junior high school has not increased significantly, which may be related to the public’s attention for the future development of this discipline. The results of each problem were discussed in detail in this article.

Keyword extraction and relevance criterion analysis based on the RAPA framework found that the Trending Keywords of information technology discipline in junior high school were Information-based teaching (r = 0.887**), Micro-lesson (r = 0.841**), Flipped classroom (r = 0.798**), Multimedia teaching (r = 0.867**), and Educational informatization (r = 0.753**). It represented the research focus of information technology curriculum in Chinese middle schools, and also reflected the diversification of teaching methods. This was similar to the views of Huang & Wang (2020), who stated that diversified teaching methods of information technology in primary and secondary schools infiltrated and crossed each other, gradually moving towards integration. This was a realistic choice to improve and innovate teaching methods. In addition, the correlation calculation results showed that the r-value between the subject network attention represented by multiple keywords and the number of academic articles was greater than 0.75, showing a positive correlation. This indicated that online media and the scientific journals had a correlation in disseminating scientific knowledge, but the timeliness of their communication content was unknown. While scholars have conducted comparative studies on this timeliness and found that scientific journals can detect hot issues in the professional field earlier than online media, and guide the public to pay attention to these issues (Li, Feng & Tian, 2017). Currently, online media and scientific journals are in parallel, which not only satisfy the public’s pursuit for distinctive content, but also drive the development of the subject.

Time series prediction based on the RAPA framework found that the attention of information technology subject in junior high school will show a flat or declining trend in the next 2 years. Moreover, the academic hot topics represented by keywords present a trend of differentiation. Information-based teaching, Multimedia teaching, and Flipped classroom are strategic ways to integrate classroom technology, improve interactivity, and enhance learning experience (Gao et al., 2020). The trend of its future attention remains the same as in recent years, indicating that the public still holds expectations for the application mode of integrating technology in future middle school information technology classroom. The attention to micro-lessons and educational informatization is showing a downward trend, especially educational informatization, whose attention index will drop to 175. Upon investigation, it was found that the future implementation of educational informatization still faces some challenges, such as issues with equipment, training, and resources. These challenges indirectly affect the attention index. In addition, regression prediction found that the number of literature in this discipline will decrease in the next few years after 2023, with an average of about 136 articles. This is consistent with Zhao et al.’s (2023) prediction trend on the literature researches of high school information technology subjects. She suggested that one of the reasons for the low number of scientific journal’s articles may be the homogenization of many literature contents. Therefore, the hot topic attention and literature researches of this discipline need to be improved.

A comparative analysis of predictions based on the RAPA framework found that the error in predicting literature researches using disciplinary network attention was relatively small. Specifically, compared with other methods such as GM(1,1), MAE in this study was reduced by 3.51 and MAPE by 1.14%. This indicates that disciplinary network attention, as an auxiliary variable in literature research, contributes to the rationality of quantitative analysis in disciplinary literature research. This was similar to the research idea of Duan & Pan (2017) and Pai, Li & Liu (2019) using real-time data of network media as intermediary variables to identify emerging disciplines. In the big data environment, network media data has strong dynamism, and the data characteristics of different disciplines are different. Therefore, the attention reflected by online media data helps researchers in different disciplines to grasp the frontier and hot spots of the discipline, and promote the progress of the discipline and science.

Impact and recommendations of the findings

The result of this study is very enlightening to the information technology subject development of in secondary school. The information technology field has always been in rapid development and transformation, and secondary school information technology subject needs to keep up with the latest technological trends in a timely manner. The results suggested that the public’s attention to the future of this discipline was not high, and the number of its literature researches was at a medium level. Information technology subject in secondary schools is a subject to train students’ practical operation and problem-solving abilities, and also an important way to cultivate their quality. Therefore, it is necessary to strengthen the development path of this discipline, such as focusing on innovation and creativity, integrating interdisciplinary knowledge, emphasizing information literacy, etc., to enhance the public’s attention to this discipline. At the same time, attention should be paid to the phenomenon of de-homogenization in literature research in this discipline, expanding the multi-dimensional research content. This is conducive to the long-term development of information technology subject in secondary schools.

In addition, the results also showed the correlation between the subject network attention and its literature researches. This reflected from the side the dynamic correlation between online platforms and scientific journals. The era of online big data has brought unprecedented challenges to traditional scientific and technological journals, changing their dissemination methods and achieving digital transformation of journals. However, their timeliness is different. The emergence of journal research hotspots often occurs earlier than the peak of online attention (Li, Feng & Tian, 2017), which helps the public better grasp the direction of online public opinion. At the same time, network attention can also help scientific journals to capture social hot issues, promoting the transformation of traditional scientific journals in the information wave. Therefore, in the process of teaching research, it is necessary to comprehensively utilize network big data and traditional data to fully reflect the more comprehensive data value, thereby improving the quality of literature research.

Research limitations and future work

This study also has some limitations. First, it only explored the correlation between discipline concern and the quantity of the academic article in the view of middle school information technology subjects. The extrapolation of research conclusions is limited. The applicability of other disciplines needs further investigation and discussion. Second, the attention data selected in this study comes from the Baidu index. Although its search engine has a great influence, there may still be statistical bias. Attention data reflects the behavior dynamics of a large number of Internet users, which may be reflected on different platforms. This article did not collect and verify data of other platforms.

Given these limitations, more comprehensive and extensive sample-size studies will be necessary in the future to validate and extend the current findings. First, the generalizability of this study’s results to other disciplines is unknown. Therefore, future studies can be conducted from disciplinary perspectives, such as English and mathematics, to explore the relationship between the attention of different disciplines and their literature researches. While expanding research findings, summarize the characteristics of different disciplines. Secondly, future research can further expand the sample types based on the data sources. For example, Altmetrics Attention Score (AAS) is a weighted metric containing online real-time data of 17 different types of network media on Altmetric.com’s comprehensive platform (Altmetric, 2024). It can measure the social and academic influence of literature research in a diversified and timely manner, making the evaluation more comprehensive, specific, and objective. In addition, 360 Trend is also a commonly used search engine (Trend, 2024). Therefore, future research can combine different indices to measure attention to compare the similarities and differences between different Index platforms.

Conclusions

Based on the constructed the RAPA framework, this article analyzed the relationship between discipline attention and its literature research from the perspective of information technology subject, and predicted the future trend of this subject. Based on the RAPA framework, this article extracted five main keywords of junior high school information technology subject: Information-based teaching, Micro-lesson, Flipped classroom, Multimedia teaching, and Educational informatization. Correlation analysis found that there was a positive correlation between this subject network attention and literature researches. Its correlation coefficient r was greater than 0.75. Predictive analysis found that public’s attention to information technology subject in junior high school is not high in the next 2 years, and its network attention shows a flat or declining trend. Meanwhile, the amount of literature has decreased compared with the previous years, with an average of about 136 articles. Prediction comparison found that the method using subject attention to predict literature researches has higher accuracy. Its MAE was 3.51 less than other methods, with a relatively small error.

The findings highlighted the complementary relationship between disciplinary attention and literature researches. Both of these are equally important to academic researchers and help them make judgments and decisions on a hot issue. At the same time, the results also indicated that the discipline needs to strengthen innovation to explore future development paths, and increase public attention and literature researches. In addition, this study also revealed the inestimable value and impact of big data represented by online attention. Therefore, while using traditional data for research, the interactivity and diversity of online media data should be considered.

Supplemental Information

Supplemental Information 1 Python code.

Supplemental Information 2 Data.

Abbreviations

COVID-19 Corona Virus Disease 2019

WoS Web of Science

LOOCV Leave-one-out cross-validation

SARIMA Seasonal AutoRegressive Integrated Moving Average

CNKI China National Knowledge Infrastructure

CJFD Chinese Journal Full-text Database

MAE Mean Absolute Error

MAPE Mean Absolute Percentage Error

AIC Akaike Information Criterion

BIC Bayesian Information Criterion

ADF Augmented Dickey-Fuller

ACF Autocorrelation Function

PACF Partial Autocorrelation Function

SD Standard Deviation

AAS Altmetrics Attention Score

Additional Information and Declarations

Competing Interests

The authors declare that they have no competing interests.

Author Contributions

Yulin Zhao conceived and designed the experiments, performed the experiments, performed the computation work, prepared figures and/or tables, and approved the final draft.

Junke Li conceived and designed the experiments, performed the experiments, analyzed the data, authored or reviewed drafts of the article, and approved the final draft.

Kai Liu analyzed the data, performed the computation work, prepared figures and/or tables, authored or reviewed drafts of the article, and approved the final draft.

Chaowang Shang performed the experiments, performed the computation work, prepared figures and/or tables, and approved the final draft.

Data Availability

The following information was supplied regarding data availability:

The raw measurements are available in the Supplemental File.

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
