# Peer review of "A hybrid integration framework based on LOOCV and SARIMA: relationship exploring and predictive analysis between discipline attention and literature research"

_PeerJ Computer Science, doi:10.7717/peerj-cs.2754_

## Round 0.1 · original submission · Major Revisions

Please revise the work carefully according to the comments. Then it will be evaluated again.

Reviewer 1 ·

Basic reporting

This study proposes a hybrid integrated method framework for exploring disciplinary relationships, and analyzes and validates it from the perspective of the information technology discipline. This is a very nice and detailed work with convincing analysis and promising results. However, there are still some issues that should be addressed. Therefore, this paper is recommended for publication after minor modification. The comments and suggestions about this work are described as follows:

1. The research gap needs to be stated. The scientific community would expect at least a non-exhaustive reference to previous studies and an analysis of their approaches to compare with the current study's methods.
2. Research contributions, which clearly describe the novelty of the proposed solution should be added in the form of bullets or numbered form (1, 2, 3, ...).
3. List all the acronyms. Authors are suggested to provide a list of acronyms and their full names before reference section.

Experimental design

1. I suggest providing a more detailed explanation of the hybrid integration framework proposed in the paper and the multiple methods used. Why use these methods? What are the advantages?
2. What does the author mean by network attention, the term network is not clear in this work. Authors are suggested to give a more detailed explanation.

Validity of the findings

1. In the "discussion" section, there is no concrete reference to the findings. The reader cannot follow the authors' rationale unless there are references to the findings. I suggest the author to correspond the specific research questions and their findings for a clearer understanding.

·

Basic reporting

By constructing a RAPA framework based on LOOCV and SARIMA, this paper analyzed the relationship between disciplinary network attention and literature research, and provided their own predictions. This paper is an interesting work and a good attempt to combine traditional CNKI data with Baidu big data. The paper contains good potential, but there are also some issues. I think the paper needs further improvement.

1.Compared with previous research methods, the highlights of this study are not yet prominent, which is the basis to understand the novelty of this paper.
2.The justification to design/choose these keywords related to educational disciplines?
3.The authors need to explain why they have considered the Baidu Index in their analysis. 
4.The RAPA method framework section should be elaborated in detail.
5.Improve the quality of Figure 5 and enlarge the font. It's blurry and small for readers to get information from the figures.
6.Paper should be checked for English spelling/minor grammatical mistakes.
7.Enrich the reference list to ensure that the literature review includes the most recent and relevant studies.

Experimental design

no comment

Validity of the findings

no comment

Reviewer 3 ·

Basic reporting

This paper discussed the development of subject teaching by considering the potential impact of big data and its attention. During the experiment, Pearson Correlation, LOOCV, SARIMA and other methods were used to test the proposed new model, and MAPE was used to verify its validity. This manuscript is well organized, and here are my specific opinions and suggestions aimed at improving the clarity, structure, and overall quality.
Introduction. A good introduction should answer the following questions: What is the problem to be solved? Are there any existing solutions? Which is the best? What is its main limitation? What do you hope to achieve? I suggest making appropriate improvements to the introduction.

Experimental design

Write a clear Methods. This section should respond to the question of how the problem was studied. Based on your paper proposing a new method, you need to include detailed information so that readers can reproduce the experiment. Perhaps you can enrich the framework of Figure 1 to improve and summarize the main phases/steps of research.

Validity of the findings

The analysis of the results is relatively comprehensive. The results are based on the application and verification of the proposed new method, and the verification process is detailed and reasonable.

Additional comments

The paper contains some complex sentences that may hinder readability. Simplify sentences where possible.
Some figures like Figure 4 present the legend over the data and it would be better to fix it.

---

## Round 0.2 · accepted · Accept

Thanks for your efforts to improve the work. This version satisfied the reviewers successfully. It may be accepted. Congrats!

Reviewer 1 ·

Basic reporting

The revised manuscript has clear logic, reasonable structure, and good innovation, and can be accepted.

Experimental design

The experimental data is sufficient and reasonable.

Validity of the findings

The conclusion of this study is very reasonable and it is recommended to accept it.

Additional comments

no comment

Reviewer 3 ·

Basic reporting

Thanks to the author for answering my doubts, I have no more questions

Experimental design

Thanks to the author for answering my doubts, I have no more questions

Validity of the findings

Thanks to the author for answering my doubts, I have no more questions